# Comparison of Antibacterial Activity of Phytochemicals against Common Foodborne Pathogens and Potential for Selection of Resistance

**DOI:** 10.3390/microorganisms11102495

**Published:** 2023-10-05

**Authors:** Ryan Sweet, Catherine Booth, Kathryn Gotts, Stephen F. Grove, Paul A. Kroon, Mark Webber

**Affiliations:** 1Quadram Institute Bioscience, Norwich Research Park, Norwich NR4 7UQ, UK; ryan.sweet@quadram.ac.uk (R.S.); catherine.booth@quadram.ac.uk (C.B.); paul.kroon@quadram.ac.uk (P.A.K.); 2McCain Foods, 1 Tower Lane, Oakbrook Terrace, IL 60181, USA; 3Norwich Medical School, University of East Anglia, Norwich Research Park, Norwich NR4 7TJ, UK

**Keywords:** antimicrobial resistance, AMR, phenolics, flavonoids, natural products

## Abstract

Antimicrobial resistance is now commonly observed in bacterial isolates from multiple settings, compromising the efficacy of current antimicrobial agents. Therefore, there is an urgent requirement for efficacious novel antimicrobials to be used as therapeutics, prophylactically or as preservatives. One promising source of novel antimicrobial chemicals is phytochemicals, which are secondary metabolites produced by plants for numerous purposes, including antimicrobial defence. In this report, we compare the bioactivity of a range of phytochemical compounds, testing their ability to directly inhibit growth or to potentiate other antimicrobials against *Salmonella enterica* Typhimurium, *Pseudomonas aeruginosa*, *Listeria monocytogenes*, and *Staphylococcus aureus*. We found that nine compounds displayed consistent bioactivity either as direct antimicrobials or as potentiators. Thymol at 0.5 mg/mL showed the greatest antimicrobial effect and significantly reduced the growth of all species, reducing viable cell populations by 66.8%, 43.2%, 29.5%, and 70.2% against *S. enterica* Typhimurium, *S. aureus*, *P. aeruginosa*, and *L. monocytogenes*, respectively. Selection of mutants with decreased susceptibility to thymol was possible for three of the pathogens, at a calculated rate of 3.77 × 10^−8^, and characterisation of *S. enterica* Typhimurium mutants showed a low-level MDR phenotype due to over-expression of the major efflux system AcrAB-TolC. These data show that phytochemicals can have strong antimicrobial activity, but emergence of resistance should be evaluated in any further development.

## 1. Introduction

Bacterial infections remain a major cause of illness and mortality for humans and animals, including those caused by foodborne pathogens ingested through contaminated foodstuffs and feedstocks [1]. This impact is being worsened by the increasing numbers of pathogenic microorganisms demonstrating antimicrobial resistance (AMR) against antibiotics used for clinical and industrial purposes [2,3,4]. In parallel with this increase in AMR, there has been a decline in the development of new antibiotics, making novel antimicrobials urgently required [5].

Plants, a historically productive source of natural pharmacological compounds, represent a source of novel biochemical structures [5], and numerous plant extracts have shown direct in vitro antimicrobial activity [6,7,8,9,10,11]. Much of this bioactivity is the result of phytochemicals, secondary metabolite compounds produced by plants for multiple functions, including antimicrobial defence against microbial pathogens [12,13,14,15]. With over 8000 compounds identified [14], phytochemicals display enormous structural diversity [15,16]. As well as the identification of direct antimicrobial activity, including carvacrol and cinnamic acid against *S.* Typhimurium and *Escherichia coli* [17] and the coumarin cajanuslactone against *S. aureus* [18], phytochemicals have also been identified that synergise with other antimicrobials via direct synergism [19,20] and the capacity to potentiate their activity via the inhibition of efflux pumps, etc. [21,22,23,24].

Despite this body of evidence demonstrating the antimicrobial properties and potential of phytochemicals, few studies have focused on responses of microorganisms to phytochemical challenge [25,26,27,28,29]. If any phytochemicals are to be developed and applied practically, a good understanding of the responses of target organisms is needed. In this study, we determined the direct inhibitive and synergistic activities of a panel of phytochemicals against standard laboratory reference strains of the common foodborne pathogens *S.* Typhimurium, *S. aureus*, *P. aeruginosa*, and *L. monocytogenes*. We then assessed the potential of the most potent compounds to select for resistant mutants, which were then characterised phenotypically and genotypically.

## 2. Materials and Methods

### 2.1. Bacterial Strains

The following bacterial strains were used: *Salmonella enterica* serovar Typhimurium 14028S, *Staphylococcus aureus* NCTC 8532, *Pseudomonas aeruginosa* PA14, and *Listeria monocytogenes* LM014. All strains were stored in Protect Microorganism Preservation System ceramic cryo-beads (Technical Service Consultants Ltd., Heywood, UK) at −70 °C until required. One cryo-bead from a stock was used to inoculate 5 mL of sterile growth medium when overnight cultures were needed; *S.* Typhimurium, *S. aureus*, and *P. aeruginosa* strains were cultured at 37 °C in Luria-Bertani Broth (LB) (Fisher Scientific BioReagents, Loughborough, UK), whereas *L. monocytogenes* was cultured at 37 °C in Brain–Heart Infusion (BHI) Broth (Oxoid, Basingstoke, UK).

### 2.2. Phytochemicals

A panel of phytochemicals consisting of caffeic acid, cinnamic acid, ferulic acid, hesperidin, kaempferol, naringenin, naringin, quercetin, rutin, thymol, vanillic acid, and vanillin was sourced from Sigma-Aldrich (Gillingham, UK) and eriodictyol from Extrasynthase (Genay, France). All compounds were stored as suggested by the manufacturer. Prosur NATPRE T-10+, chosen because it is a commercially available phytochemical mixture designed to substitute sodium nitrite/nitrate as a food preservative, was stored in aluminium foil-wrapped Durans, sealed with parafilm, and kept in a dark environment. All phytochemicals were dissolved at room temperature in DMSO, vortexed, and stored at 4 °C. Compounds were selected based on the rationale of balancing previously described activity and the general cost of synthesis/purchase. Compounds representative of most phytochemical structural families were selected. Tested phytochemical concentrations were selected to reflect the working concentrations of the commercially available Prosur NATPRE T-10+ mixture and not those found in natural extracts.

### 2.3. Growth Inhibition Assays

A high-throughput screen was used to test the panel of all phytochemicals for antibacterial activity. From 1 mg/mL phytochemical working stocks, dilutions to give final concentrations of 0.5 mg/mL, 0.25 mg/mL, and 0.125 mg/mL were made and 100 µL of each were added to three wells of 96-well microtiter plates. Plates were then inoculated with bacteria diluted in 100 µL of medium (LB for *S.* Typhimurium, *S. aureus*, *P. aeruginosa* strains; BHI for *L. monocytogenes*) at ~10^−5^ CFU/mL. Drug-free media controls were included as well as broth containing DMSO at the final concentrations used as solvent controls. Plates were sealed with gas-permeable membranes (ThermoFisher Scientific, Cambridge, UK) and incubated at 37 °C overnight for 16 h. After incubation, the OD_(600 nm)_ of each culture was measured using a FLUOstar Omega plate reader (BMG Labtech, Ortenberg, Germany). Endpoint OD_(600 nm)_ readings were used instead of a time-course experiment to allow for a more rapid screening of multiple plates at once. All experiments were replicated four times and statistically analysed as detailed below (see Section 2.10).

### 2.4. Identification of Antimicrobial Potentiation

The ability of each phytochemical to synergise with chloramphenicol (chosen as an antibiotic whose activity is increased by agents that enhance membrane permeability or inhibit efflux) was determined. Plates were inoculated with phytochemicals as described in Section 2.3 above, but the media used to inoculate bacteria contained chloramphenicol at a final concentration of 2 µg/mL. Plates were incubated and analysed as for direct growth inhibition assays; phenyl-arginine β-naphthylamide (PAβN) was included as a control, being a known efflux pump inhibitor. All experiments were replicated four times and analysed as detailed below. Phytochemicals that displayed significant reduction in OD_(600 nm)_ measurements with the inclusion of chloramphenicol, but not in its absence (at least not to the same extent), were deemed as holding the capacity for antibiotic potentiation.

### 2.5. Measuring Cell Viability after Phytochemical Exposure

To distinguish bacteriostatic and bactericidal activity, the viable numbers of cells after exposure to a panel of selected phytochemicals (chosen based on results from the high throughput end-point assay above) were determined. Overnight bacterial cultures were set up in triplicate for each species. The next day, fresh growth media was supplemented with phytochemicals at either 0.05 mg/mL or 0.5 mg/mL and inoculated with ~10^5^ CFU/mL of each species (overnight bacterial cultures were diluted appropriately into 250 µL of PBS). These cultures were then sampled by removing 20 µL at 0, 1, 2, 4, 8, and 16 h of incubation at 37 °C and ~200 rpm. Each aliquot was serially diluted and 20 µL spots inoculated onto LB agar plates and incubated overnight at 37 °C before colony-forming units (CFU) were enumerated. In total, three technical replicates for each of three biological replicates were tested in this way and analysed as detailed below.

### 2.6. Drug Accumulation Assays

To determine whether any phytochemicals impacted cellular permeability, the accumulation of resazurin, a molecule that is metabolised within the cell into a fluorescent product, was measured. Intracellular accumulation of resazurin is usually limited due to low permeability, and it is also a substrate for efflux pumps. Accumulation was measured in the presence and absence of the selected phytochemicals. Overnight bacterial cultures were grown, then used to incubate fresh growth media the following day. Once bacterial cultures had reached an OD_(600 nm)_ of 0.2–0.5, cells were harvested and resuspended in sterile PBS, normalising the OD_(600 nm)_ to the lowest measurement taken. A 96-well plate was inoculated with 5 µL of a 400 µg/mL resazurin stock (20 μL of a 100 μM ethidium bromide stock was substituted for *S. aureus* and *L. monocytogenes*), as well as 2 µL of sterile PBS/DMSO/50 mg/mL phytochemical stock depending on the experimental condition. Positive controls included an additional 5 µL of a 5 mg/mL PAβN stock and, finally, 193 µL of bacterial suspensions were added to the appropriate wells and mixed. In total, five technical replicates for each of three biological replicates were tested in this way. As soon as all wells were filled, a transparent membrane was sealed onto the 96-well plate, which was placed into a FLUOstar Omega Plate Reader to measure OD_(600 nm)_ and fluorescence readings (excitation: 544 nm, emission: 590 nm) over a period of 16 h. 

### 2.7. Mutant Selection

Phytochemical-laced agar was inoculated with either *S.* Typhimurium, *S. aureus, P. aeruginosa*, or *L. monocytogenes* and incubated to select for resistant mutants. Muller Hinton agar plates were prepared and supplemented with either caffeic acid, thymol, or the Prosur NATPRE T-10+ mix at 1× and 2× the MIC. These phytochemicals were selected due to their potent directly antimicrobial activity, their capacity for the potentiation of chloramphenicol, or their current use as a food preservative substitute. Supplemented plates were then inoculated with bacteria concentrated to 10^−10^ or 10^−9^ CFU in 100 µL volumes and incubated at 37 °C overnight. A dense inoculum was used as the classic methodology to encourage and capture the potential for mutant selection from exposure to the tested compounds/mixture. Control plates with no phytochemicals or with DMSO as a vehicle control were prepared in parallel. The next day, the plates were enumerated, the average mutation frequency was calculated (see below), and random mutant colonies were picked via toothpick before culturing and storage in 40% glycerol stocks at −70 °C. Plates that showed no colony growth were incubated for a further 24 h, then re-examined. 

### 2.8. Mutant Sequencing and Phenotyping

Mutant strains of *S.* Typhimurium selected by phytochemical exposure were cultured for DNA extraction and sequencing as described recently [30]. Briefly, cultured cells were lysed and DNA was isolated via DNA-binding magnetic beads (KAPA Pure Beads, Roche Diagnostics) and eluted with 10 mM Tris-Cl, pH 8.5. The extracted DNA samples were sequenced using an Illumina Nextseq500 instrument [30] and sequence reads were quality filtered using Trimmomatic (v3.5) with default parameters. Contigs were assembled using SPAdes v3.11.1 and the de novo and parental genomes fed through the Snippy v3.1 software to identify single nucleotide polymorphisms (SNPs) [30]. The sequenced mutant strains were also phenotypically assayed using the growth analysis and drug accumulation assays described above, and the MICs of antibiotics kanamycin, tetracycline, ampicillin, chloramphenicol, and nalidixic acid were also determined via the microdilution broth method. Finally, to identify whether mutant strains had any difference in biofilm-forming ability, crystal violet assays and analysis of colony morphology on congo red plates were used, both as described previously [31]. 

### 2.9. Transmission Electron Microscopy (TEM)

An overnight culture of *S.* Typhimurium and a previously selected thymol-tolerant mutant strain were used to inoculate 200 mL LB within Erlenmeyer flasks in a 1:200 ratio. After an overnight incubation at 37 °C and ~200 rpm, these flask cultures were separated into 30 mL aliquots within 50 mL centrifuge tubes and supplemented to contain 0.25–1 mg/mL thymol, with PBS and DMSO controls included. These samples were incubated at room temperature for two hours before being pelleted, washed in 10 mL PBS, and pelleted again to be covered with 1 mL PBS. The pelleted samples were then processed by the QIB Bioimaging Core Facility for sample fixing and TEM imaging. Briefly, 2.5% glutaraldehyde in 0.1 M sodium cacodylate buffer (both Agar Scientific Ltd., Stansted, UK) was used as the fixative, post fixed in 1% osmium tetroxide (Agar Scientific Ltd., UK) for two hours, and dehydrated through an ethanol series (30–90% for 15 min each, followed by 100% ethanol three times each for 15 min). Finally, samples were embedded in resin through a series mix of LR White medium-grade resin (Agar Scientific Ltd., Stansted UK) and 100% ethanol (1:1, 2:1, 3:1 resin:ethanol, and, finally, three times 100% resin, each for at least 1 h with the final 100% resin step overnight), before polymerising overnight at 60 °C and sectioning the samples to ~90 nm thickness using an ultramicrotome (Leica EM UC6, Wetzlar, Germany) onto carbon-coated copper TEM grids (EM Resolutions Ltd., Keele, UK) and sequential staining with ~2% uranyl acetate (BDH 10288) and 0.5% lead citrate-tribasic trihydrate (Sigma 153265-25G). Sections were examined and imaged in an FEITalos F200C transmission electron microscope at 200 kV with a Gatan One View digital camera. Digital micrograph files (.DM4) were converted to a .TIFF format for ease of viewing. The resulting images were parsed to identify representative examples of the bacterial morphologies and analysed as described below.

### 2.10. Statistical Analyses

For all of the previously described experiments, the following statistical analyses were implemented. For the growth inhibition assays and identification of antimicrobial potentiation, all experiments were statistically analysed via a one-way ANOVA test with Fischer’s least significant difference post-test. For measuring cell viability after phytochemical exposure and the drug accumulation assays, growth/accumulation velocities and endpoint states were first calculated. The data were then analysed using a one-way ANOVA test with the inclusion of Fischer’s least significant difference post-test. For the mutant selections, the following formula was used to calculate a mutation frequency for each pathogen against each tested phytochemical at 1 × MIC.
$$Average\;mutation\;frequency=\frac{\left[Average\;CFU/mL\;for\;each\;inoculum\right]}{Corresponding\;DMSO\;viable\;count\;(CFU/mL)}$$

For the phenotyping of thymol-selected mutants, the previously stated analyses for the measuring of cell viability after phytochemical exposure and the drug accumulation assays were performed. Crystal violet assays were analysed using a one-way ANOVA test with the inclusion of Fischer’s least significant difference post-test, whereas congo red assays were quantitatively observed. Finally, five individual cells randomly selected from the TEM images were analysed via the ImageJ version 1.53r software (National Institutes of Health, MD, USA). Mean grey values were measured. Mean grey values were labelled as average cytoplasmic density values in the corresponding table, as this was the intended inference. Numerical values were displayed to two decimal places and a statistical analysis via a one-way repeated measures ANOVA test, with the inclusion of Fisher’s LSD test, performed using the GraphPad software package version 8.0, to distinguish statistically significant results.

## 3. Results

### 3.1. Nine Compounds Show Consistent Antimicrobial Activity

All 14 compounds were screened for both direct antimicrobial activity and the ability to potentiate other antimicrobial agents, with most demonstrating activity in either assay. Examples of results from growth inhibition and potentiation assays are shown in Figure 1a,b, with full data in the Appendix (Table A1, Table A2, Table A3 and Table A4). Nine compounds showed a reproducible, statistically significant reduction in the average OD_(600 nm)_ achieved by cultures of the four tested pathogens, although the spectrum of activity varied between compounds. Dose-dependent inhibition of growth was also observed; for example, the flavanone eriodictyol reduced the average OD_(600 nm)_ measurements of *S. aureus* and *L. monocytogenes* cultures by 86.87% (at 0.5 mg/mL) and 12.31% (at 0.125 mg/mL), respectively, compared to the relevant controls (Figure 1a,b). Naringenin showed potent activity against *S. aureus* (reducing growth by 92.8%, 89.5%, and 8.9% at 0.5 mg/mL, 0.25 mg/mL, and 0.125 mg/mL, respectively). Thymol at 0.5 mg/mL showed a wide spectrum of activity and reduced the average OD_(600 nm)_ of *S.* Typhimurium, *P. aeruginosa*, and *L. monocytogenes* by 48.82%, 42.59%, and 23.35%, respectively, although it was not active against *S. aureus*. 

The data points show the final OD achieved after 16 h of incubation of four biological replicates, with three technical replicates each. The horizontal bars show the mean for each set. The dashed line shows the OD from media alone. Statistical analysis was performed using the GraphPad software v.8, using a one-way ANOVA test with Fischer’s least significant difference test. The error bars indicate SEM (±). 

PABN, a known efflux inhibitor, was used as a control to potentiate the activity of chloramphenicol (present at 0.25 × MIC in all conditions).

To determine impacts on viability, the number of surviving cells was enumerated after exposure to two concentrations of the compounds that demonstrated activity in the initial optical density screens. Eight of the nine tested compounds exerted a significant reduction in viability numbers of at least one pathogen (Figure 2 and Figure A1 and Table A5, Table A6, Table A7 and Table A8). 

In general, the phytochemicals were more effective at inhibiting the growth of the Gram-positive than the Gram-negative pathogens tested, and dose-dependent effects were again observed. 

Prosur NATPRE T-10+ significantly reduced the CFU/mL of *L. monocytogenes* cultures by 38.8% compared to the relevant control; however, this compound mixture did not have a significant inhibitive effect on any other tested pathogen (see Figure 2). Eriodictyol significantly decreased the growth velocity and endpoint states of *S. aureus* by 84.7% and 44.1%, respectively, and of *L. monocytogenes* by 61.2% and 35.3%, respectively (Figure 3). The flavonols quercetin and kaempferol both significantly decreased the growth velocities of *S. aureus* (quercetin and kaempferol) and *L. monocytogenes* (quercetin alone) cultures (Figure A1 and Table A5, Table A6, Table A7 and Table A8). The flavanone naringenin showed a statistically significant reduction in growth of all four pathogens (Figure 2 and Figure A1 and Table A5, Table A6, Table A7 and Table A8). Thymol was significantly more active than any other compound and exerted the most consistent and potent inhibitive activity against all of the tested pathogens (Figure 2). This was particularly pronounced for earlier time points with decreased growth velocities for *S.* Typhimurium (70.2% reduction), *S. aureus* (75.9% reduction), *P. aeruginosa* (87.6% reduction), and *L. monocytogenes* (>99% reduction). The final number of viable cells achieved by each population was also lower after thymol exposure for *S.* Typhimurium (66.8% reduction), *S. aureus* (43.2% reduction), *P. aeruginosa* (29.5% reduction), and *L. monocytogenes* (70.2% reduction) compared to the appropriate controls. 

The Prosur NATPRE T-10+ mix and vanillin exhibited dose-dependent effects on *L. monocytogenes*, with the former displaying inhibitive activity at 0.5 mg/mL but not at the lower concentration of 0.05 mg/mL and vice versa for the latter aldehyde compound (see Table A8 and Table A9).

### 3.2. Caffeic Acid and Prosur NATPRE T-10+ Enhance Cellular Permeability

Drug accumulation after exposure to the phytochemicals was determined, and selected results are shown in Figure 3 (full data can be found in Figure A2, Figure A3, Figure A4 and Figure A5 and Table A10, Table A11, Table A12 and Table A13). Six compounds provoked a statistically significant increase in drug accumulation by *S.* Typhimurium, *P. aeruginosa*, and *L. monocytogenes* cultures. Caffeic acid (0.5 mg/mL) increased the velocity of resazurin accumulation within *S.* Typhimurium by 40.8% compared to the solvent vehicle control and provoked a smaller increase in *P. aeruginosa* cultures (Figure A2, Figure A3, Figure A4 and Figure A5, Table A10, Table A11, Table A12 and Table A13). The Prosur NATPRE T-10+ mixture was most effective at increasing the drug accumulation of all the tested pathogens, increasing the rate of resazurin accumulation by between 70 and 200% in *S.* Typhimurium (Figure 3) and *P. aeruginosa* (Figure A2, Figure A3, Figure A4 and Figure A5, Table A10, Table A11, Table A12 and Table A13).

Ferulic acid and naringenin were also able to increase resazurin accumulation, increasing the average fluorescence accumulation velocity of *S.* Typhimurium by 29.1% and 17.9%, respectively (Figure 3). For *S. aureus* and *L. monocytogenes* (Figure 3), ethidium bromide was used rather than resazurin as the fluorescent indicator due to these microorganisms’ rapid metabolisation of the latter compound. This showed increased accumulation after exposure to the Prosur NATPRE T-10+ mix in S. aureus (Figure 3). 

### 3.3. Thymol Selects for Resistant Mutants at a Rate Similar to Classic Antibiotics

To determine the potential for resistance emergence, *S.* Typhimurium, *S. aureus*, *P. aeruginosa*, and *L. monocytogenes* were exposed to the MIC and 2× the MIC of caffeic acid, Prosur NATPRE T-10+, or thymol in agar. After inoculation of plates with large populations of each organism, no significant inhibition was observed from caffeic acid or the Prosur NATPRE T-10+ mix, and bacterial lawns of growth were observed. Thymol, however, was able to inhibit growth of the majority of populations and selected for discrete mutant colonies of *S.* Typhimurium, *S. aureus*, and *P. aeruginosa* at an average rate of 3.77 × 10^−8^ (Table 1).

### 3.4. Thymol-Selected Mutants of S. Typhimurium Display Tolerance Rather Than Resistance to Thymol

Eight randomly selected *S.* Typhimurium mutant colonies were characterised for thymol susceptibility; no strain grew at or above the MIC of 0.5 mg/mL, but at 0.25 mg/mL thymol the candidate mutant strains produced larger colonies than the parental strain (Figure A6). 

### 3.5. S. Typhimurium Mutant Sequencing Reveal Efflux-Associated SNPs

The mutant strains of *S.* Typhimurium were sequenced, and a total of four SNPs were identified. *S.* Typhimurium mutant strain #2 was found to harbour a substitution within the putative class I SAM-dependent methyltransferase yafS gene, causing a H107Q substitution within the protein. This strain #2 also possessed a unique substitution within the tetR/acrA family transcriptional regulator ramR gene, resulting in the substitution of phenylalanine amino acid 48 of RamR to cysteine, a change not currently identified within the literature. *S.* Typhimurium mutant strain #1 contained another known mutation within ramR, resulting in a G96D amino acid residue substitution. 

One final SNP identified within three mutants (#1, #5, and #6) was the insertion of a thymine base between STM14_18795 (a putative cytoplasmic protein, glpF homologue) and STM14_18790 encoding the putative DeoR family transcriptional regulator glpR. 

### 3.6. Thymol-Tolerant S. Typhimurium Mutants Display Decreased Susceptibility to Antibiotics

As sequencing of the thymol-selected mutant strains revealed the presence of SNPs within loci associated with efflux activity, the antibiotic sensitivity of the strains was determined (Table 2). Mutant strains #1 and #2 demonstrated small increases in the MICs of tetracycline, ampicillin, chloramphenicol, and nalidixic acid, all antibiotics subject to efflux, which is consistent with the ramR mutations seen in these strains. 

Values show the average MIC fold changes for five antibiotic compounds (Kan = kanamycin, Tet = tetracycline, Amp = ampicillin, Chl = chloramphenicol, and Nal = nalidixic acid) against mutants of *S.* Typhimurium compared to the parental (WT) strain. Data are the average of three biological replicates (each with three technical replicates).

### 3.7. S. Typhimurium Mutant Growth in the Presence of Thymol

*S.* Typhimurium mutant strains #1 and #2 were further investigated via growth curves to quantify their growth kinetics under 0.25 mg/mL thymol challenge; the results of these experiments are presented in Figure 4. Thymol at 0.25 mg/mL reduced the growth of both the mutant strains and the parental *S.* Typhimurium strain, whereas all cultures grew with no significant differences if unchallenged (Figure A7). No statistically significant difference was seen between the mutants and the parent in growth in the presence of thymol.

### 3.8. S. Typhimurium Mutants Accumulate Less Resazurin Then Parent Strain

Figure 4 depicts the accumulation of resazurin by selected *S.* Typhimurium strains and shows lower accumulations for both the ramR mutant *S.* Typhimurium strains #1 and #2 compared to the parental control (statistically significant for mutant #2 only). This is consistent with increased efflux activity.

### 3.9. S. Typhimurium Mutants Display a Decreased Biofilm Capacity

Crystal violet staining assays were performed to determine the thymol-tolerant mutants’ biofilm-forming capacity. Figure 4 shows the biomass of biofilms produced by the parent and mutants. The thymol-tolerant *S.* Typhimurium mutants exhibited decreased biomass production compared to the parental strain. This trend was further observed in congo red plating, with a lesser degree of staining for the resulting colonies (Figure A8).

### 3.10. Thymol Exposure Induces Damage to the Bacterial Envelope

Transmission electron microscopy (TEM) was used to observe the morphological effects of thymol exposure on bacterial cells of *S.* Typhimurium. Figure 4 presents the morphologies of the parental and mutant #2 strains with and without exposure to 1 mg/mL thymol. The parental strain presented a classic envelope and bacillary morphology when unchallenged with thymol. After 2 h of exposure, the parental *S.* Typhimurium strain presented a significantly altered rod shape with a decreased cytoplasmic density and envelope thickness and a “fluffy” cellular surface (Figure 4, Table 3). In contrast, the efflux-associated mutant *S.* Typhimurium strain #2 presented a more rounded morphology with a thinner envelope and rougher surface even under control conditions. However, this morphology was largely unaltered when exposed to 1 mg/mL thymol (Figure 4, Table 3).

Quantitative values produced through analysis of five randomly selected cells across two TEM images are presented above. Average cytoplasmic density relates to mean grey values; the higher the value, the less dense the cytoplasmic contents of the analysed cells. Values presented at two decimal places. Statistical analysis was performed using GraphPad software v.8 using a one-way repeated measures ANOVA Test with Fischer’s LSD test. Values in bold are statistically significant compared to the relevant control. 

## 4. Discussion

The initial panel of 14 phytochemicals screened for antimicrobial activity against foodborne pathogens was chosen considering previous reports of activity and availability; of these, nine showed some activity. Previous work has suggested hesperidin [32], rutin [33], and vanillic acid [34] to have antimicrobial activity, but this was not observed at any tested concentration against any pathogen tested here. Of the nine phytochemicals selected for further testing, seven displayed a capacity to inhibit the growth of at least one microorganism tested. Naringenin at a concentration of 0.5 mg/mL did not present significant activity against *S.* Typhimiurium, corroborating other works that present an MIC of 1 mg/mL [35], although it did exert a significant reduction in OD_(600 nm)_ measurements against *S. aureus*, adding to earlier work in the field that has found this to be the compounds’ MIC [36]. Quercetin, although unsuitable for the OD-based assays devised here due to its natural yellow pigmentation, has previous evidence supporting its antimicrobial capacity [37] and was thus carried forward for further investigation. Kaempferol was also selected on a similar basis [38]. Other compounds from the selected nine have also had previous studies reporting on their antimicrobial activity, such as eriodictyol [39]. Thymol, however, was the most potent phytochemical and showed consistent bioactivity against the panel of organisms, which supports previous studies reporting MIC ranges of 0.005–0.662 mg/mL against these organisms [40,41,42,43,44]. Alternative evidence supporting the antimicrobial activity of thymol suggests an MIC of 0.25 mg/mL against *S. aureus* and a minimum bactericidal concentration (MBC) of 0.5 mg/mL [45]. In contrast we observed an increase in CFU/mL from under the detection threshold after eight hours of incubation in the presented growth curves, suggesting that, although 0.5 mg/mL may be the MIC, the MBC under these conditions may in fact be higher due to various environmental, methodological, and genetic factors. Research to improve the solubility and stability of antimicrobial phytochemicals, including the nanocomplexation and production of thymol composites, have also shown an increase in antimicrobial activity against many foodborne pathogens [45,46].

In addition to direct antibacterial activity, there is potential for phytochemicals to be deployed in combination with other antimicrobials [47,48,49,50]. Although checkerboard assays have traditionally been utilised for this end and remain a promising methodological avenue for future research, drug accumulation assays were used to infer synergistic capacities via the phytochemical’s action of permeabilising the bacterial cell within the present work. Four of the selected nine phytochemicals (caffeic acid, ferulic acid, naringenin, and vanillin) displayed an ability to increase accumulation of resazurin/ethidium bromide, suggesting applications as potentiating agents [23,49,51,52,53,54,55,56,57,58,59]. Caffeic acid’s potential to synergise with conventional antimicrobials has been described previously against *E. coli*, *P. aeruginosa*, and *S. aureus* strains [47,60,61]. We extend these data to now include *S.* Typhimurium. In contrast, researchers have published data providing evidence that thymol possesses synergistic effects [42,62,63,64], including Miladi H. et al. (2016) [65], who employed a similar methodology in the ethidium bromide accumulation assay. The authors found that thymol (in addition to the related compound carvacrol) at 0.5 mg/mL inhibited the efflux of ethidium bromide from cells of *S. aureus*, *E. coli*, *S.* Typhimurium, and *Salmonella entiriditis* [65]. We did not predict this conclusion from our data, potentially because thymol was found to be rapidly bactericidal in our experiments, making synergies hard to detect. In addition, there are differences in the employed methodologies; we used drug accumulation assays to directly identify compounds that were likely to impact membrane permeability or inhibit efflux. Other work has relied on checkerboard assays, which identify synergies but provide no mechanistic information. 

Few studies to date have investigated the potential of phytochemicals to select for resistant populations. To explore this, we determined the frequency of mutant selection for *S.* Typhimurium, *S. aureus*, *P. aeruginosa*, and *L. monocytogenes* when exposed to caffeic acid, thymol, and the Prosur NATPRE T-10+ mix. Of these, caffeic acid and the Prosur NATPRE T-10+ mix were unable to inhibit bacterial growth on agar plates when challenged with the large inocula used for mutant selection. Thymol, however, did select for mutant colonies at an average rate of 3.77 × 10^−8^ across the four pathogens tested. This mutation frequency was similar (approximately ~10^−8^ to 10^−9^ [66,67,68]) to that found for multiple bacterial species, including the common foodborne pathogens tested here, when challenged with classical antibiotics such as quinolones [66], fluoroquinolones [68], and norfloxacin [67].

The *S.* Typhimurium mutants demonstrated a mild level of tolerance to thymol itself, with no MIC change, but evidenced improved growth in the presence of thymol. Two of the mutants also showed decreased susceptibility to antibiotics (tetracycline, ampicillin, chloramphenicol, and nalidixic acid), reduced accumulation of the efflux substrate resazurin, and lower biofilm biomass production (Figure 4). Sequencing of the thymol-selected *S.* Typhimurium mutants revealed the presence of SNPs within the *yafS* gene [69,70], not characterised in detail for *S.* Typhimurium, and upstream of *glpR*, a repressor of the sugar/carbohydrate transport and metabolization *glp* operon. Perhaps more relevant, however, were the multiple SNPs within the AcrAB efflux pump transcriptional repressor *ramR* [71] identified within different mutants. RamR is a well-known repressor of RamA, which in turn controls expression of many genes, including the *acrAB* multidrug efflux system [72]. Similar work with *E. coli* [73] selected a thymol-resistant strain (JM109-Thyr) via repeated exposure to sub-lethal thymol concentrations. Resulting isolates demonstrated two-fold increased thymol MICs and had mutations in *acrR*, the local repressor of *acrAB* [73]. Our data further suggest that thymol can select for mutants with decreased susceptibility and that they are efflux over-expression mutants (via *ramR* rather than *acrR* in *Salmonella*). These mutants demonstrate a classic efflux phenotype of low-level multidrug resistance, further supported by the decreased capacity for biofilm production presented in this work, which may have implications for the selection of collateral resistance following thymol exposure. This potential for the selection of phytochemical and cross-resistance in bacterial pathogens is an important line of scientific enquiry with consequences for the practical adoption of phytochemicals as substitute antimicrobials in the clinic, farm, household, or food industries. The latter is particularly true given the push for non-chemical food preservatives [74,75] and the status of phytochemicals as generally recognised as safe (GRAS) for inclusion in food products [76]. If antimicrobial phytochemicals are to be employed en masse, the possibility of bacterial resistance to these and extraneous compounds evolving along the same lines as the current AMR crisis must be considered.

Finally, the identification of efflux-associated SNPs within the thymol-selected mutants prompted the application of TEM to observe the effect of thymol challenge on the cell envelope morphology of *S.* Typhimurium. Here, we observed significant envelope damage and cytoplasmic shrinkage of the parental strain, with a greater robustness exhibited by the thymol-tolerant mutant when exposed to thymol (Figure 4, Table 3). This disruption of the bacterial envelope/membrane is in line with other studies that observed similar results both in *S. enterica* and in other bacterial species [77,78,79,80,81] and supports previous suggestions for interactions with the bacterial envelope as mediating the antimicrobial mechanism of action for thymol. Although there is a strong evidence base to support an envelope-targeting mechanism for thymol, the TEM imaging implemented within this study only provides a semi-qualitative observation. Future work may focus on the use of scanning electron microscopy to better observe the envelope surface and assays designed to investigate the impact of the observed effects on cell viability and cellular functions. There have also been, however, multiple mechanisms of antimicrobial action reported for this monoterpenoid phenol. There are numerous reports supporting the inhibition of efflux pumps, the disruption of natural biofilm functions, the inhibition of motility and key bacterial enzymes [82], and the dysregulation of protein and DNA synthesis [75]. This phenomenon of multiple inhibitive mechanisms is not unique to thymol, as it appears that many phytochemicals share a multi-pronged paradigm of antibacterial activity [75].

## 5. Conclusions

We have shown that various phytochemicals have antimicrobial effects against important foodborne pathogens, that this differs between compounds and species, and that some phytochemicals also demonstrate good promise as potentiators/synergisers in combination with other antimicrobial agents. Thymol was the most active compound tested, and although we were able to select for mutants with decreased susceptibility to thymol, they were efflux mutants, which has implications for thymol application, as these mutants also displayed a decreased susceptibility to antibiotics. Although phytochemicals represent an alternative source of directly and synergistic antimicrobial compounds to be exploited, further research is needed into their mechanisms of action and potential for selection of resistance.

## Figures and Tables

**Figure 1 microorganisms-11-02495-f001:**
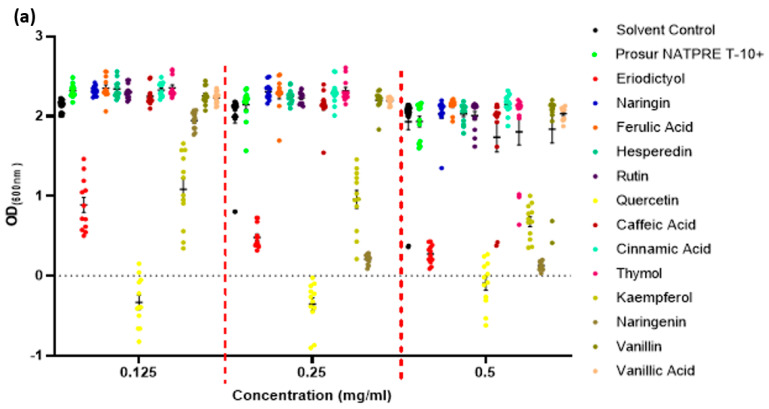
Example antibacterial and potentiating activity of phytochemicals. (**a**) Growth of *S. aureus* in the presence of different phytochemicals. (**b**) Impact of phytochemicals on growth of *S.* Typhimurium in the presence of a sub-MIC concentration of chloramphenicol. Red dotted lines included to visually segregate the tested concentrations.

**Figure 2 microorganisms-11-02495-f002:**
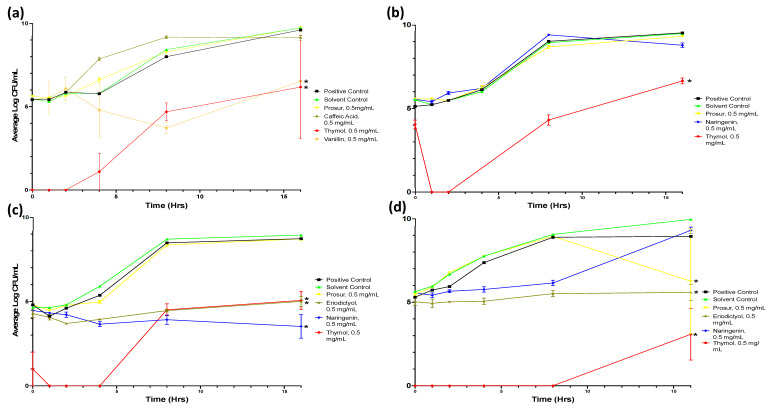
Viability of pathogens challenged with 0.5 mg/mL of selected phytochemicals. Viable numbers (based on CFU/mL) of the following: (**a**) *S.* Typhimurium, (**b**) *P. aeruginosa*, (**c**) *S. aureus*, and (**d**) *L. monocytogenes* following exposure to different phytochemicals. Experiments were repeated with three biological replicates (three technical replicates each) over an incubation period of 16 h. Graphs display the averaged values of three technical replicates for three biological replicates. Error bars indicate SEM (±); asterisks denote statistical significance.

**Figure 3 microorganisms-11-02495-f003:**
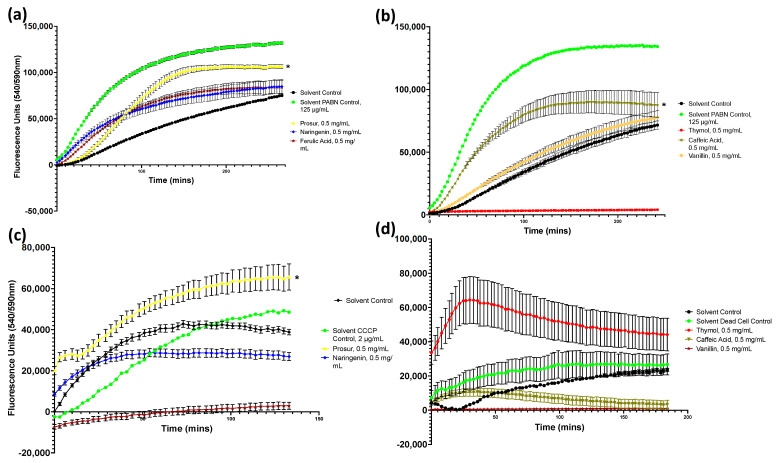
Selected resazurin accumulation assays for pathogens challenged with 0.5 mg/mL of various phytochemicals. (**a**) *S.* Typhimurium challenged with the Prosur NATPRE-T10+ mix, naringenin, and ferulic acid. (**b**) *S.* Typhimurium challenged with thymol, caffeic acid, and vanillin at 0.5 mg/mL. the Prosur NATPRE-T10+ mix, naringenin, and ferulic acid. (**c**) *S. aureus* challenged with the Prosur NATPRE-T10+ mix, naringenin, and ferulic acid. (**d**) *L. monocytogenes* challenged with thymol, caffeic acid, and vanillin. Points show the blank-adjusted and averaged values of five technical replicates for three biological replicates. Error bars indicate SEM (±); asterisks denote statistical significance.

**Figure 4 microorganisms-11-02495-f004:**
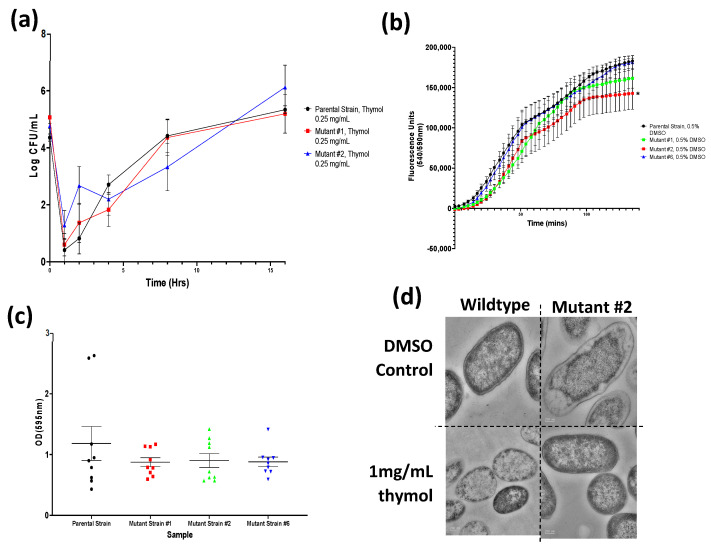
Characterisation of thymol-tolerant *S.* Typhimurium mutants. (**a**) Growth of parent and mutant strains in the presence of thymol (data show averages from nine replicates); asterisks denote statistical significance. (**b**) Resazurin accumulation assays (points show average values from five technical replicates of three biological replicates); asterisks denote statistical significance. (**c**) Biofilm biomass measured by crystal violet assays (points show average values from nine replicates). (**d**) *S. enterica* wild-type and mutant #2 TEM images under thymol and equivalent solvent vehicle exposure.

**Table 1 microorganisms-11-02495-t001:** Frequency of selection of thymol-tolerant mutants.

Microorganism	Frequency of Mutant Selection
*S.* Typhimurium	6.84 × 10^−9^
*S. aureus*	1.47 × 10^−7^
*P. aeruginosa*	3.01 × 10^−8^
*L. monocytogenes*	N/D *
Average for all species	3.77 × 10^−8^

* N/D: not determined due to an absence of identifiable colonies.

**Table 2 microorganisms-11-02495-t002:** Changes in antibiotic susceptibility of thymol-selected *S.* Typhimurium mutants. Parental strain MICs for the listed antibiotics, in order as presented, are as follows: 2.67 μg/mL, 1 μg/mL, 6 μg/mL, 3.33 μg/mL, and 4 μg/mL. A # denotes the allocated number of the mutant strain.

Average MIC Fold Changes of Mutant Strains Relative to Parent						
		Antibiotic				
		Kan	Tet	Amp	Chl	Nal
*S.* Typhimurium	WT	1.00	1.00	1.00	1.00	1.00
	#1	1.00	2.67	4.21	3.42	4.17
	#2	0.83	2.67	4.21	3.42	4.17
	#6	1.00	1.67	2.38	1.67	2.00

**Table 3 microorganisms-11-02495-t003:** Quantitative ImageJ analysis of TEM images from thymol-exposed *S.* Typhimurium. Values in bold denote statistically significant samples.

*S.* Typhimurium TEM Image Analysis			
Sample	Average Cytoplasmic Density	SEM (±)	*p*-Value
**WT control**	128.2	1.92	-
**WT solvent control**	**98.6**	**8.66**	**0.0045**
**WT 1 mg/mL thymol**	140.5	5.96	0.1986
***S. enterica*** **#2 control**	145.5	2.66	-
***S. enterica*** **#2 solvent Control**	139.8	1.90	0.0912
***S. enterica*** **#2 1 mg/mL Thymol**	**128.5**	**1.26**	**<0.0001**

## Data Availability

Data is contained within the article.

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
