# Peer review of "Comparison of Antibacterial Activity of Phytochemicals against Common Foodborne Pathogens and Potential for Selection of Resistance"

_microorganisms, 2023, doi:10.3390/microorganisms11102495_

Round 1
Reviewer 1 Report
Overall the manuscript is well written and well presented. Graphical presentation attract the readers attention in reading this manuscript. The manuscript draft has sufficient data to be published.
Manuscript draft improvement is required at few places and I am mentioning a few but the author should thoroughly check for the typos.
Author should read and correct Page 3 line 97 and 98 "The next day fresh growth media ....". It should be improved.
Page 3 line 98 "~105 CFU/ml" Kindly confirm.
Discussion section: 2nd paragraph line 345 starting with "as well as direct antibacterial activity....". Replace as well as with a more appropriate word.
Author Response
Referee 1 Comments and Actions
Overall the manuscript is well written and well presented. Graphical presentation attract the readers attention in reading this manuscript. The manuscript draft has sufficient data to be published.
Manuscript draft improvement is required at few places and I am mentioning a few but the author should thoroughly check for the typos.
Manuscript was thoroughly checked and identified typos were corrected.
Author should read and correct Page 3 line 97 and 98 "The next day fresh growth media ....". It should be improved.
This sentence was improved and now reads “The next day, fresh growth media was supplemented with phytochemicals at either 0.05mg/mL or 0.5mg/mL and inoculated with ~105 CFU/mL of each species (overnight bacterial cultures were diluted appropriately into 250µL of PBS).” on Lines 161-164.
Page 3 line 98 "~105 CFU/ml" Kindly confirm.
Inoculum was of a standard density of ~105 CFU/ml. Text was changed to include the appropriate superscript and now reads “~105 CFU/ml”, Line 162.
Discussion section: 2nd paragraph line 345 starting with "as well as direct antibacterial activity....". Replace as well as with a more appropriate word.
“as well” has been replaced with the more appropriate “In addition to...”. The 2nd paragraph of the Discussion section starts on Line 649, with Figures and Tables in situ.
Reviewer 2 Report
Manuscript 2565603
Journal Microorganisms
Title Comparison of antibacterial activity of phytochemicals against common foodborne pathogens and potential for selection of resistance
The manuscript entitled “Comparison of antibacterial activity of phytochemicals against common foodborne pathogens and potential for selection of resistance” describes the antibacterial action of 13 phytochemicals against 4 foodborne pathogens, their capacity to boost the antibacterial activity of antibiotics and the emergence of mutant strains resistant to these phytochemicals. The manuscript is interesting but several parts need improvement. Please follow the comments in the file.

English is fine. Some sentences could be improved.
Author Response
Referee 2 Comments and Actions
Manuscript 2565603 Journal Microorganisms
Title Comparison of antibacterial activity of phytochemicals against common foodborne pathogens and potential for selection of resistance
The manuscript entitled “Comparison of antibacterial activity of phytochemicals against common foodborne pathogens and potential for selection of resistance” describes the antibacterial action of 13 phytochemicals against 4 foodborne pathogens, their capacity to boost the antibacterial activity of antibiotics and the emergence of mutant strains resistant to these phytochemicals. The manuscript is interesting but several parts need improvement. Please follow the comments below:
L11-25 Please revise the abstract including quantitative data
The abstract has been revised to include quantitative data
L40-41 Please introduce the plant antibacterial compounds (e.g., phenolic compounds, terpenes and so on), their efficacy against foodborne pathogens, and their use as boosters of common antibiotics. Please use relevant references
We have expanded some of the introduction as suggested: “As well as identification of direct antimicrobial activity, including carvacrol and cinammic acid against S. Typhimurium and Escherichia coli (E. coli)[17] and the coumarin cajanuslactone against S. aureus[18], phytochemicals have also been identified which synergize with other antimicrobials via direct synergism[19, 20] and the capacity to potentiate their activity via the inhibition of efflux pumps, etc.[21-24].”
L47 Please better describe this part. Synergistic activity with antibiotics? Authors did not evaluate the synergistic activity...synergistic action can be evaluated through the determination of the FICI values and the MIC values of each compound and of selected combinations. Revise
We have been clearer about how we tested for synergy via potentiation of an efflux substrate here and the text is now more specific. We have added some more suitable references f; Lines 64-67.
Reference #17, Yoon, B.I., et al., Anti-inflammatory and Antimicrobial Effects of Anthocyanin Extracted from Black Soy-bean on Chronic Bacterial Prostatitis Rat Model. Chinese Journal of Integrative Medicine, 2018. 24(8): p. 621-626., does specify a synergistic effect with ciprofloxacin so we believe it is a relevant citation.
Reference #18 Seukep, Armel Jackson, et al. "Plant-derived secondary metabolites as the main source of efflux pump inhibitors and methods for identification." Journal of pharmaceutical analysis 10.4 (2020): 277-290 has been shifted to better reflect the cited authors’ writings and the referee’s comments.
Reference #19 Wolska, Krystyna I., Katarzyna Grzes, and AnnA KuREK. "Synergy between novel antimicrobials and conventional antibiotics or bacteriocins." Pol J Microbiol 61.2 (2012): 95-104. directly states and details synergy between novel antimicrobials and conventional antibiotics, and so the authors believe it is a relevant citation.
#Reference 20 Adu, F., et al., Antibacterial resistance modulatory properties of selected medicinal plants from Ghana. African Journal of Pharmacy and Pharmacology, 2019. 13(5): p. 57-69. has been shifted to better reflect the cited authors’ writings and the referee’s comments.
L54-56 Are these strains multi-drug resistant? Why these strains were selected? Please explain
No, these strains are typical laboratory reference strains and the MS has been amended to reflect this in Lines 71-74; “In this study we determined the direct inhibitive and synergistic activities of a panel of phytochemicals against standard laboratory reference strains of the common foodborne pathogens S. enterica serovar Typhimurium (S. Typhimurium), S. aureus, P. aeruginosa and L. monocytogenes.”. Future work could and should use multi-drug resistant strains, however this is beyond the scope of this study.
L64-67 Why these phytochemicals are selected? Do they belong to different chemical classes? Please explain in the text
See response to Referee 3; Lines 97-102.
L67-69 Why NATPRE T-10 was used? It is a mixture of phytochemicals whereas other phytochemicals are pure compounds. Please explain this choice.
Prosur NATPRE T-10+ was chosen as it is a commercially available phytochemical mixture designed as a substitute food preservative to replace sodium nitrite/sodium nitrate. This explanation is included in Lines 92-95; “Prosur NATPRE T-10+, chosen as it is a commercially available phytochemical mixture designed to substitute sodium nitrite/nitrate as a food preservative, was stored in aluminum foil wrapped Durans, sealed with parafilm and kept in a dark environment.”.
L71-83 Why 100 μL of phytochemicals dissolved in DMSO was used? Why 10^5 cfu mL-1 was used? How 0.5 mg/mL was selected? Why only end-point measurement was made? Please revise this section addressing these questions, also with relevant references
We have added more detail of the methods and Lines 105-107 have been amended to reflect this.
An inoculum of 10-5 cfu was used to allow potential for bacterial growth over a 3 log range and therefore capability to measure inhibition of polyphenols.
See response to referee 3 re concentrations used
Endpoint OD(600nm) measurements were taken as this allowed higher throughput and the main aim was to screen for compounds which could provoke and maintain a reduction on viable numbers over an extended time (informing later more detailed time course measurements)
L81-83 Please move this part in a new statistical section
A separate Materials and Methods subsection, Statistical Analyses, has been added to the MS to describe the analyses for all included experiments. See Lines 295-320: “Statistical Analyses
L83 Probably post-hoc test.
The statistics section has been created describing tests used
L84-91 How the potentiation effect was measured? It is unclear. Please better explain in this section.
This has been expanded for clarity, Lines 154-156: “Phytochemicals that displayed significant reduction in OD(600nm) measurements with the inclusion of chloramphenicol, but not in its’ absence (at least not to the same extent), were deemed as holding the capacity for antibiotic potentiation”.
L91-92 Please move this part in a new statistical section
See above
L98 How these concentrations were selected? Please explain in the text
See response to referee 3
L98-99 Please better describe the preparation of inoculum
See response to referee 1
L99-102 How plate count was performed using only 20 μL of samples? Please better describe in the text
Apologies for the vagueness, to clarify- The 20mL aliquots were serially diluted down the rows of a 96-well microtiter plate filled with 180mL PBS. From these dilutions, 5mL was spotted onto LB agar within square petri dishes, and CFU were enumerated. Lines 164-166 were amended to reflect this; “These cultures were then sampled by removing 20µL at 0,1, 2, 4, 8 and 16 hours of incubation at 37°C and ~200 rpm. Each aliquot was serially diluted in 180µL of PBS and colony forming units (CFU) enumerated after 5µL samples were spotted and incubated on LB agar plates overnight at 37°C.”.
L102-105 Please move this part in a new statistical section
Done
L111 Which strains were used? Please include this information
Added to the methods
L114 Lowest OD value? The OD value should be equal for all strains. Please explain this choice
Apologies, the ODs were equalised but within an accepted (small range) – within that samples were equalised to the lowest OD
L116 50 mg/mL phytochemicals? Please check
Yes, 50mg/mL was used as the phytochemical stock for these experiments to enable a smaller volume to be added to the wells of the 96-well microtiter plate, reducing the DMSO content of the samples while still allowing for the concentrations tested.
L122-124 Please move this part in a new statistical section
Done
L128-129 Why only three compounds/mixture were selected? Why 1010-109 cfu/mL? Please explain in the text
Three compounds were selected based on highest potency in the initial screen, and high inoculums were used to ensure as many potential mutants within a population which could be assayed were present as is typical for mutant selection. This has been made clear now in the text (Lines 236-249)
L132-133 How the mutation frequency was measured? Please better describe this part
Lines 245-247 amended to clarify: “The next day the plates were enumerated, the average mutation frequency was calculated (see below), and random mutant colonies were picked via toothpick before culturing and storage in 40% glycerol stocks at -70°C.”.
L137-138 Please describe the genome/DNA analysis and the identification of SNPs. Materials and Methods of this part is lacking
The referenced paper now details the methodologies for the genome/DNA analysis and SNP identification (Lines 252, 255, 258, 264). This section has been updated with a very brief description of the procedures, however (Lines 251-264): Mutant strains of S. Typhimurium selected by phytochemical exposure were cultured for DNA extraction and sequencing as described recently[30]. Briefly, cultured cells were lysed, DNA was isolated via DNA-binding magnetic beads (KAPA Pure Beads, Roche Diagnostics) and eluted with 10mM Tris-Cl, pH 8.5. The extracted DNA samples were sequenced using an Illumina Nextseq500 instrument[30] and sequence reads were quality filtered using Trimmomatic (v3.5) with default parameters. Contigs were assembled using SPAdes v3.11.1 and the de novo and parental genomes fed through the Snippy v3.1 software to identify single nucleotide polymorphisms (SNPs)[30]. The sequenced mutant strains were also phenotypically assayed using the growth analysis and drug accumulation assays described previously and the MICs of antibiotics kanamycin, tetracycline, ampicillin, chloramphenicol and nalidixic acid were also determined via the microdilution broth method. Finally, to identify whether mutant strains had any difference in biofilm forming ability crystal violet assays and analysis of colony morphology on congo red plates were used, both as described previously[31].”
L140-144 Please better describe these assays. Revise this part
The assays are previously described earlier in the Materials and Methods section, and within the referenced publication. The text has been modified: Lines 258-264: The sequenced mutant strains were also phenotypically assayed using the growth analysis and drug accumulation assays described previously and the MICs of antibiotics kanamycin, tetracycline, ampicillin, chloramphenicol and nalidixic acid were also de-termined via the microdilution broth method. Finally, to identify whether mutant strains had any difference in biofilm forming ability crystal violet assays and analysis of colony morphology on congo red plates were used, both as described previously[31].
L150-163 Please rewrite this part. It is difficult to follow this procedure. Please use short sentences
We have tried to break up the sentences in this section
L171-174 Please move this part in a new statistical section
A separate Materials and Methods subsection, Statistical Analysis, has been added to the MS to describe the analyses for all included experiments. See Lines 295-320.
L174-175 Please add a new statistical section
A separate Materials and Methods subsection, Statistical Analysis, has been added to the MS to describe the analyses for all included experiments. See Lines 295-320.
L176-225 Please rewrite this part. It is confusing. Please re-organize this section. The antibacterial activity in the end-point assay should be presented at the begin of the section against all the pathogens. Then, the effect on the growth rate (not velocity!!) should be presented. Revise
The antibacterial activity in the end-point assays are briefly presented in the Results section first (Lines 294-299), and we have made the rational for mentioning velocity clearer as well as pointing to the appendix with all the data gathered from these experiments.
L176-225 Here and throughout the manuscript, please use italics for the species of foodborne pathogens
The MS has been reviewed and this issue has been solved.
Figure 2 Why different number of phytochemicals was evaluated against the 4 pathogens? Please explain in the text
Showing all the data in the main text is impractical and we have made the legend clearer to indicate the selection displayed with additional data presented in the Appendix.
Figure 2 Please indicate significant differences with different letters and include standard deviation in all growth curves. Revise
All Figures have been edited for clarity, indicators of significance and the addition of standard deviation where possible.
Figure 3 Data are not easy to understand. Please use figures in the Fig. A2-A5 (relevant figures for each pathogen) to replace Figure 3 with a new figure reporting the main relevant data. Please use a good resolution
The Figures have been modified to be clearer according to other referee comments
L226-242 Please use data of Tables A10-A11 and Tables A12-A13 to summarize the results of accumulation test on Gram-positive and Gram-negative bacteria. These four tables are helpful for the interpretation of the results.
See above.
Table 1 How the frequency of isolation was calculated? Please explain
This issue has been addressed earlier in the list of Referee 2’s comments.
L261-262 Which is the rate of classic antibiotics? Please add this information with relevant references
Amended, these rates are referenced directly in the Discussion section (Lines 774-779): “Thymol however did select for mutant colonies at an average rate of 3.77x10-8 across the four pathogens tested. This mutation frequency is similar (approximately ~10-8 to 10-9[66-68]) to that found for multiple bacterial species, including the common foodborne pathogens tested here, when challenged with classical antibiotics such as quinolones[66], fluoroquinolones[68] and norfloxacin[67].”
L265-270 Please move this part in the section 3.3
Table has been placed according to the Journal’s guidelines of placing Figures/Tables directly after their first citation within the main text.
L271-282 Which are the implications of these efflux-associated SNPs? Please better explain the results in this section
A fuller description of the affected genes and SNPs is included in the Discussion section (Lines 780-807).
Table 2 The inclusion of MIC values of WT strain is strongly suggested
WT strain’s MIC values have been included in the Table 2 legend, Lines 554-556; “Table 2. Changes in antibiotic susceptibility of thymol-selected S. Typhimurium mutants. Wildtype MICs for the listed antibiotics, in order as presented, are as follows: 2.67mg/mL, 1mg/mL, 6mg/mL, 3.33mg/mL and 4mg/mL.”.
L294-300 Please include the reduction of cell viability after 1 h of exposure to thymol and the growth until the end of incubation. Results of different assays should be better described by the authors. Revise
No time-course data exists for this experiment as it was a standard microbroth dilution MIC assay, utilising endpoint OD(600nm) measurements. Section now on Lines 780-807.
L301-305 and Figure 4 panel B Is this reduction significant? Please include statistics in the text at L301-305 Figure 4C Is this reduction significant? Please include statistics in the text at L306-311
Results which are significant are described within the MS text. Figures have been amended to highlight statistically significant data with an asterisk.
L317-334 Please better explain the results of TEM images. In particular, which are the implications of increase/decrease of cytoplasmatic density on cell morphology under control or thymol treatment? Please better describe these results. Actually, this part is confusing and not clear
The TEM images are a qualitative observation (Lines 598-608). The discussion has been altered for clarity (Lines 816-820): “Although there is a strong evidence base to support an envelope-targeting mechanism for thymol, the TEM imaging implemented within this study only provides a semi-qualitative observation. Future work may focus on the use of scanning electron microscopy to better observe the envelope surface, and assays designed to investigate the impact of the observed effects on cell viability and cellular functions.”
L340-342 Please better discuss this part including references on the antibacterial activity of these phytochemicals and their MIC values
The authors have edited the Discussion section to include references to the antimicrobial activity of select members of the phytochemical panel tested in Figures 1a and 1b, Lines 653-662. As this screen is not the main focus of the MS, a full discussion seems tangential to the authors.
L344-345 The different antibacterial activity of thymol against Gram-positive and Gram-negative bacteria as well as the species more resistant to thymol exposure should be aspects for the discussion. The papers doi.org/10.1016/j.foodchem.2020.127594, doi.org/10.1016/j.heliyon.2022.e09551, and doi.org/10.1016/j.micpath.2016.08.008 are suggested for your analysis and discussion.
The authors thank the referee for these relevant papers and have edited the MS to include these in the Discussion section (Lines 662-673).
L354-359 Please revise this part. Synergistic action should be evaluated by combining the checkerboard assay with other tests (e.g., accumulation assay). In order to find a synergistic action thymol should be diluted and a low concentration should be tested. The checkerboard assay is based on the comparison between the MIC values of single compounds and the MIC values of combinations. Please revise this part considering these aspects. Both methodologies should be used to evaluate synergistic actions. Please include this information in the discussion section
As detailed in an earlier comment, this MS uses a specific measure of potentiation focused on efflux inhibition and the MS has been edited to clarify this (Lines 675-695).
L384-386 The paper doi.org/10.1016/j.micpath.2016.08.008 could be discussed in the part related to the efflux-pump inhibition displayed by thymol against some foodborne pathogens
The authors thank the referee for this relevant paper and have edited the MS to include this in the Discussion section (Lines 679-695).
L335-398 A paragraph discussing the mode of action of thymol against foodborne pathogens and, generally bacteria, is necessary. The papers doi.org/10.1080/10408398.2019.1675585 and doi.org/10.3390/foods12122315 are suggested for your analysis and discussion.
The authors thank the referee for these relevant papers and have edited the MS to include these in the Discussion section (Lines 817-848).
L399-408 Please do not refer to synergistic activity/synergism. Please add more details related to your data and implications for the use of phytochemicals against foodborne pathogens and the emergence of possible mutant strains
The authors have edited the Discussion section to reflect the issues of arising phytochemical resistance in bacterial pathogens (Lines 799-807) and is already highlighted in the Conclusions section (Lines 852-857).
References Please revise the reference list including the suggested papers and revising the numbering in the text
Done
Reviewer 3 Report
The MS deals with a screening of several phytochemical for their antimicrobial activity. The MS also gives insights about the emergence of antibiotic resistance. The topic is crucial in this years and the data collected interesting. However, some issues in the presentation of the data must be fixed prior to publication.
I suggest to merge Results and Discussion, to make more readable the MS.
How did the Authors choose the studied phytochemicals? The choice must be justified and explained in the MS to understand the logic of the work. I suggest to include the chemical formula of the compounds and some considerations about their phytochemistry.
Similarly, the concentrations used should be explained, since they appeared quite high and not often realistic in natural extracts.
Figure 1: it is not easy to gain information from this graph. Maybe the Authors could try to further splitting the image or somehow make it more clear to the reader.
Similarly, even the other figures have the same problem. The Authors collected a huge amount of data, but they could be presented in a more effective way to the readers.
In this perspective, splitting Results and Discussion did not help.
It is preferable to use “mL” or “uL” (capitol L) instead of “ml”.
Line 184 and over: scientific names should be italicized all over the MS.
Author Response
Referee 3 Comments and Actions
The MS deals with a screening of several phytochemical for their antimicrobial activity. The MS also gives insights about the emergence of antibiotic resistance. The topic is crucial in this years and the data collected interesting. However, some issues in the presentation of the data must be fixed prior to publication.
I suggest to merge Results and Discussion, to make more readable the MS.
This suggestion has been considered, however we feel it is clearer to delineate the results and discussion sections and also follows the journals formatting guidelines. We are happy to follow editorial advice though if this is a preferred and allowable change.
How did the Authors choose the studied phytochemicals? The choice must be justified and explained in the MS to understand the logic of the work. I suggest to include the chemical formula of the compounds and some considerations about their phytochemistry.
Phytochemicals were chosen from an in house library, based on a balance of described activity, chemical structure and synthesis/purchasing costs. This rationale has been added to Lines 97-102; “Compounds were selected based on the rationale of balancing previously described activity and the general cost of synthesis/purchase. Compounds representative of most phytochemical structural families were selected.
Phytochemical concentrations tested were selected to reflect those in commercial application in the available Prosur NATPRE T-10+ mixture.” Lines 97-102
Similarly, the concentrations used should be explained, since they appeared quite high and not often realistic in natural extracts.
See previous comment
Figure 1: it is not easy to gain information from this graph. Maybe the Authors could try to further splitting the image or somehow make it more clear to the reader.
We have split Figure 1a and 1b across two separate pages (Lines 361-403).
Similarly, even the other figures have the same problem. The Authors collected a huge amount of data, but they could be presented in a more effective way to the readers.
We have modified the figures throughout the MS to improve clarity.
In this perspective, splitting Results and Discussion did not help.
See previous comment
It is preferable to use “mL” or “uL” (capitol L) instead of “ml”.
The text has been amended.
Line 184 and over: scientific names should be italicized all over the MS.
Scientific names have been checked throughout the MS and have been italicized. “Typhimurium”, denoting a serovar and due to convention, is not italicized.
Round 2
Reviewer 2 Report
Auhtors addressed reviewer's comments. Minor changes are suggested below:
L119-122 Please revise this part adding the right volume used to inoculate plates
Minor editing changes are necessary
Author Response
Thank you for the review - the volumes for colony enumeration have been added to the text as requested
Reviewer 3 Report
The Authors effectively managed to improve the MS.
However, I am still convinced that merging the two sections would have improved the MS.
Author Response
We would like to thank the referee again for careful consideration of our manuscript and we acknowledge the suggestion re merging the results and discussion. However we believe the journal format guidelines prefer separate sections so have left this as it is - we will however of course defer to the editors guidance and judgement on this point